# Comparison of Accumulated Degree-Days and Entomological Approaches in Post Mortem Interval Estimation

**DOI:** 10.3390/insects12030264

**Published:** 2021-03-21

**Authors:** Lorenzo Franceschetti, Jennifer Pradelli, Fabiola Tuccia, Giorgia Giordani, Cristina Cattaneo, Stefano Vanin

**Affiliations:** 1Forensic Medicine Unit, Department of Medical and Surgical Specialties, Radiological Sciences and Public Health, University of Brescia, Piazzale Spedali Civili, 1, 25123 Brescia, Italy; lorenzofranceschetti@gmail.com; 2LABANOF (Laboratorio di Antropologia e Odontologia Forense), Sezione di Medicina Legale Dipartimento di Scienze Biomediche per la Salute, Università degli Studi di Milano, Milano Via Luigi Mangiagalli, 37, 20133 Milano, Italy; cristina.cattaneo@unimi.it; 3School of Applied Sciences, University of Huddersfield, Queensgate, Huddersfield HD1 3DH, UK; jennifer.pradelli@gmail.com (J.P.); tuccia.fabiola@gmail.com (F.T.); 4Dipartimento di Farmacia e Biotecnologie (FABIT), Alma Mater Studiorum Università di Bologna, 40126 Bologna, Italy; giorgia.giordani.gg@gmail.com; 5Dipartimento di Scienze della Terra dell’Ambiente e della Vita (DISTAV), Università di Genova, Corso Europa 26, 16132 Genova, Italy; 6National Research Council, Institute for the Study of Anthropic Impact and Sustainability in the Marine Environment (CNR-IAS), Via de Marini 6, 16149 Genova, Italy

**Keywords:** ADD, TBS, Diptera, PMI, colonisation, temperature

## Abstract

**Simple Summary:**

Among the investigative questions to define the temporal frame of a criminal event, the time since death plays a fundamental role. After death, the body goes through a series of physical and chemical transformations—known as decomposition. The way in which different parts of the body undergo these transformations can be quantified with a scale of scores (TBS, the total body score) and used for the time since death evaluation using the accumulated degree-days (ADDs) parameter, which accounts for time and temperature. This method is reported as TBS/ADD. Another way to estimate the time since death is based on the insect development on the body. Flies represent the first body coloniser and the development of their immature stages is used to define the time of colonisation that is temperature dependent and species specific. In this study, the two methods were compared based on 30 forensic cases occurring in northern Italy. The results highlighted the limits of the TBS/ADD method and the importance of the entomological approach, keeping in mind that with insects the colonization time is evaluated. This time is the minimum time since death.

**Abstract:**

Establishing the post mortem interval (PMI) is a key component of every medicolegal death investigation. Several methods based on different approaches have been suggested to perform this estimation. Among them, two methods based their evaluation on the effect of the temperature and time on the considered parameters: total body score (TBS)/accumulated degree-days (ADDs) and insect development. In this work, the two methods were compared using the results of minPMI and PMI estimates of 30 forensic cases occurring in northern Italy. Species in the family Calliphoridae (*Lucilia sericata*, *Calliphora vomitoria* and *Chrysomya albiceps)* were considered in the analyses. The results highlighted the limits of the TBS/ADD method and the importance of the entomological approach, keeping in mind that the minPMI is evaluated. Due to the fact that the majority of the cases occurred in indoor conditions, further research must also be conducted on the different taxa to verify the possibility of increasing the accuracy of the minPIM estimation based on the entomological approach.

## 1. Introduction

Establishing the post mortem interval (PMI) is a key component of every medicolegal death investigation. The longer the time since death, the more imprecise the chronological or sequential indicators detectable on the remains are [1]. A reliable PMI estimation is extremely important for the reconstruction of events surrounding the death. Testimonial statements provided by neighbours, family members and co-workers are not always reliable, thus, crime investigators normally base the PMI assessment on ante mortem physical evidence when available (e.g., video recording, cell phone records, bank activities, calendars, social networking, medicine reminders, medical records, etc.). However, scientifically, the most accurate and precise PMI estimate relies on post mortem changes and decomposition processes, which are the prerogative of forensic pathologists or, depending on the country, of other forensic medical specialists (e.g., coroner) [2].

Human decomposition is a complex biological and chemical process that begins immediately after death and involves the interaction of cadaver enzymes, bacteria, fungi, and protozoa [3,4]. Body decomposition is characterised by stages of gradual physical decay, from the fresh stage to skeletonization, through a bloated stage, followed by an active and then an advanced stage [5,6,7]. Vass (2011) described the four main abiotic factors that affect the decomposition rate: temperature, moisture, pH and the partial pressure of oxygen [8]. In addition, other parameters influence the decomposition such as the cause of death, wounds or trauma, bodyweight, degree of exposure to sunlight, body coverage, insect activity and subsoil parameters [9,10,11,12,13,14,15].

Temperature is the most important factor affecting decomposition and for this reason, meteorological information is fundamental to estimate the PMI [9]. Obtaining measurements with the best possible confidence of the actual temperatures the body experienced at the potential crime scene is crucial for an accurate estimation [16]. After death, the environmental temperature affects the body’s chemical and biochemical processes, impacting the decomposition and at the same time, affecting the entomological colonisation, both in terms of body search and insect development. Moreover, the temperature and moisture also have great influence on other organisms that can develop on or around the body, such as plants, fungi, bacteria and other microorganisms [17].

In forensic practice, methods for PMI estimation are based on the macroscopic examination of the soft tissues’ decomposition degree. However, when the skeletal stage is reached, few markers for this purpose exist [18,19]. In the literature, the morphological changes that take place during decomposition have been described in detail [5,20]. Most of these descriptions are qualitative, based on personal opinions and experience, and not applicable to all geographic and environmental conditions [10,21,22]. Furthermore, many decomposition studies have been conducted in different seasons and climatic conditions using varying methods [15,22,23]. To bypass the limitation mentioned above, a method based on the observation of decomposition stages of different body districts have been developed in forensic anthropology. The body decomposition degree can be quantified using the total body score (TBS) [10]. It is a scale that distinguishes the different stages of decomposition, allowing to assign points to specific categories and eventually to score overall decomposition [24,25]. Numerous studies utilised the TBS method either with or without some modification [8,24,25,26,27,28,29,30,31,32].

Due to the relationship between temperature and decomposition, a semi-quantitative model to estimate PMI was based on accumulated degree-days (ADD). ADD represent the combination between chronological time and temperature. They are defined as heat-energy units representing the accumulation of thermal energy needed for chemical and biological reactions to take place in soft tissue during decomposition [10]. This method takes into consideration the overall body decomposition (evaluated by a score like the TBS) and the number of environmental degrees recorded since the death. Megyesi et al. (2005) proposed a retrospective study in which TBS and ADD could be used to quantitatively estimate PMI [10]. A few authors have tested Megyesi’s research in their countries and have found positive correlation [33,34,35]. However, worldwide there is an increasing number of authors that did not manage to corroborate Megyesi’s equation at their latitude (such as South Africa [22], Netherlands [32], Australia [36], Canada [37]). An alternative method for PMI estimation is based on the study of the insects developing on a body, taking in account that insect development is temperature dependent and species specific [38,39,40,41,42]. Forensic entomology uses, depending on the case, the development of necrophagous insects and the composition of insect communities present on the body after the death for the minimum PMI (minPMI) estimation defined as the colonisation time [41,42,43,44,45,46,47,48,49,50,51].

Despite several attempts and new technologies, medicolegal death investigation still lacks a universal and reliable method to estimate PMI that allows an investigation to proceed appropriately and without delay, while providing time for more complex analyses [2]. Currently, there is no scientifically recognised PMI estimation method for any specific or general geographic region. As more and more knowledge is gained from thanatological experimental studies, a working model that encompasses the factors affecting decomposition is becoming more and more plausible. Furthermore, since the environment plays a large role in the rate of decomposition, the applicability of current PMI models needs to be tested and validated at a regional level.

For these reasons, this study was aimed to test the efficiency and reliability of the semi-quantitative method of TBS and ADD applying the Megyesi mathematical formula [10] on real forensic cases and to compare it with the entomological evaluations of the PMI to verify the complementarity of the two methods in death investigations.

## 2. Materials and Methods

### 2.1. Accumulated Degree-Days Analysis

Thirty human corpses in an advanced decomposition stage from the Lombardia region (Northern Italy), subjected to on-site external examination and then to autopsy at the Institute of Legal Medicine, University of Milan, between 2016 and 2018, were included in this study. The bodies were stored under refrigerated conditions in adherence to local laws between the scene recovery and the autopsy. Photographs shot during the crime scene inspection and during the autopsy were used to quantify/evaluate the level of decomposition of each body using the TBS as suggested by Galloway et al. (1989) [5] and Nawrocka et al. (2016) [24]. The ADD for the PMI estimation were calculated using Megyesi’s formula [10].

### 2.2. Entomological Analysis

The insects were sampled from the bodies before their removal from the crime scene and later during the autopsy following the EAFE (European Association for Forensic Entomology) guidelines [51] and stored at −20 °C until analysis as per GIEF (Gruppo Italiano di Entomologia Forense) protocol (GIEF, 2016) [52]. After defrosting, the larvae were fixed with hot water (80 °C, 1 min) and finally stored in an 80% ethanol solution.

Entomological samples were observed and photographed using a Leica M60 stereomicroscope equipped with a DFC425C camera and the LAS software (Leica, Germany).

Species were identified using morphological keys [53,54] and confirmed by molecular analysis after dissection as described by Tuccia et al. (2016) [55]. DNA was extracted from a fraction of larval tissue using the QIAamp DNA Investigator Kit (QIAGEN, Redwood City, CA, USA). The manufacture protocol was partially modified in order to increase the quality of the reaction by adding 4 μL of RNAse A (4 mg/mL) after over-night incubation with the Proteinase K (100 μg/mL) (PROMEGA, Madison, WI, USA). DNA was eluted in 100 μL of elution buffer and quantified via Qubit 3.0 fluorimeter (Thermo Scientific, Waltham, MA, USA). Polymerase Chain Reaction was carried out on mitochondrial COI gene 658 bp long and commonly used as a molecular target for insects’ barcoding [56]. Universal LCO-1490 Forward primer (5′-GGTCAACAAATCATAAAGATATTGG-3′) and HCO-2198 Reverse primer (5′-TAAACTTCAGGGTGACCAAAAAATCA-3′) were used as described by Folmer and co-workers (1994) [57]. PROMEGA GoTaq^®^ Flexi Polymerase protocol (PROMEGA, Madison, WI, USA) was followed in order to prepare a master mix reaction of 20 μL final volume: 4 μL of Colourless GoTaq Flexi Buffer (5×), 2 μL of MgCl_2_ (25 mM), 0.5 μL of each primer (10 pmol/μL), 0.5 μL of Nucleotide Mix (10 mM), 0.25 μL GoTaq DNA Polymerase (5 u/μL) and 2 μL of DNA template. The following amplification program was set up on BioRad C1000 Thermal Cycler (Bio-Rad Laboratories, Inc., Hercules, California, USA): 95 °C for 10 min, 35 cycles of 95 °C for 1 min, 49.8 °C for 1 min, 72 °C for 1 min and a final elongation step at 72 °C for 1 min. Each reaction was confirmed by standard gel electrophoresis in 1.5% agarose gel previously stained with Midori Green Advanced DNA Stain (Geneflow, Elmhurst, UK). Thirty-five microliters of PCR products was purified using QIAquick PCR Purification Kit^®^ (QIAGEN, Hilden, Germany) following the manufacturer’s instructions. Purified amplicons were eluted in 40 μL of sterile/deionised water and sequenced by an external company (Eurofins Operon MWG, Ebersberg, Germany).

### 2.3. PMI Estimation and Statistical Analysis

Insect development is temperature dependent and species/population specific in the range between the minimal and the maximal developmental thresholds. Depending on the nature of the sample (already fixed or living specimens, larvae, close or empty puparia) two different methods can be applied to estimate the age of an insect. ADD is currently the most common method used for this estimation when living specimens can breed until the adult stage or when empty puparia are collected from the crime scene. Because of the minimal developmental threshold, the ADDs used in entomology differ from the ADDs applied in the anthropological method. For this reason, ADDs were derived from the average temperature as absolute values for the morphological evaluation, whereas if needed for the entomological evaluation, the minimal developmental threshold was subtracted from the absolute value.

minPMI was also evaluated using the larvae measurements applying the data available in the literature [58] when the samples were already fixed. In this case, the size of the larvae was compared with diagrams reporting the relations between size, temperature and time. Whereas some authors only report the size as a function of the temperature and time [58,59,60], others also provide the formulas and the interval of confidence of the larval length based on the ADDs [61]. Larvae were counted and then measured using Leica M60 stereomicroscope equipped with a CCD camera (Leica, Wetzlar, Germany) and the automatic measurement tool. When the numbers of larvae were under fifty, measurements were performed on all the specimens. When the count exceeded fifty, only fifty larvae were considered. Measurements were expressed in mm as the average and standard deviation (SD).

The estimations derived from the two methods were compared using an interclass correlation according to Cicchetti (1994) [62] and Koo and Li (2016) [63]. Statistical analyses were performed using the SPSS Statistics software (IBM, Armonk, NY, USA).

## 3. Results

Twenty-four of the deceased cases were males (80%) whereas 6 were females (20%). At the time of death, the mean age was 63 years (range of 37–88 years). According to the autopsy’s reports, the main cause of death was cardiovascular diseases (22 cases, 73.5%), followed by cerebrovascular diseases and acute drugs intoxication (three cases, 10%). Pneumoniae was the cause of death in two cases (6.5%). The last time the person was seen alive ranged between 2 and 10 days. The average temperature for the period before body recovery is summarised per each case in Table 1.

### 3.1. Accumulated Degree-Days Estimate

The decomposition scores were evaluated for every region, and then added up to calculate the TBS. Data are summarised in Table 1, whereas the derived ADD with the PMI based on circumstantial data and entomological data are reported in Table 2. One case was removed from the PMI evaluation because the body conditions did not allow the application of the morphological method.

The PMI based on the ADD estimates in 12 cases (40%) was in good agreement with the circumstantial data; in the remaining cases, it was overestimated (15 cases, 50%) or underestimated (three cases, 10%).

All cases resulting as positive outcomes were above 5 days for PMI, with the exception of one case, which had a PMI estimate in 4 days (Table 2).

### 3.2. Entomological Estimate

Calliphoridae sampled from the cases and used for the estimation were: *Lucilia sericata* (Meigen, 1826) in 16 cases (53%), *Calliphora vicina* Robineau-Desvoidy, 1830 in 11 cases (37%) and *Chrysomya albiceps* (Wiedemann, 1819) in three cases (10%).

In 21 cases (70%), the minimum entomological PMI correlates positively with the time of death from the circumstantial data; in the other cases there is an underestimation (9–30%) of the PMI provided by circumstantial information (Table 2).

### 3.3. Comparison

In 10 cases (33%), both methods allowed to have an estimate of the post-mortal interval in agreement to what was given by the circumstantial data. In 9 out of the 10 cases, the PMI exceeded over 5 days. In the tenth case, the PMI was included between 4 and 5 days (Table 2).

### 3.4. Statistical Analysis

Interclass correlation for the absolute value based on the central tendency value which resulted to be 0.58 (95% CI: 0.11–0.80) for circumstantial data vs. the ADD estimate and 0.71 for circumstantial data vs. the entomological evaluation (95% CI: 0.39–0.86).

Due to the fact that the ADD method estimates the PMI whereas the entomological one estimates the minimum PMI, the interclass correlation of these two variables was estimated in terms of consistency and not, as in the previous analysis, in terms of absolute value. The results indicate a value around 0.65–0.69 (95% CI: 0.25–0.85) for the lower, upper and central tendency values of the estimates, with an average underestimation of 49 h (95% CI: 36.4–61.5) for the lower limit and of 19.7 h (95% CI: 1.1–38.2) for the upper limits of the estimate.

## 4. Discussion

In this study, the PMI provided by the circumstantial data showed high variability, from a minimum of two days to a maximum of ten days. The ADD estimates did not allow to positively correlate the derived TBS with the circumstantial data in 60% of the cases. In these cases, the TBS scale presented a higher score of decomposition for the head/neck region, compared to the other body regions. These cases are distributed throughout the whole year and in no case was the body naked, the only parts exposed to the air were the head and the extremities of the limbs. In the remaining cases (40%), a positive concordance between the ADD estimate and the PMI derived from circumstantial data was found. The positive outcomes were shown for PMI above 5 days. This concordance can be explained by the fact that the ADD conversions and subsequent estimations became more accurate during the later stages of decomposition, as suggested by Parsons (2009) [34].

For shorter intervals, the ADD derived from the TBS scale lost effectiveness, while forensic entomology remains the most reliable method. This aspect should be considered as an important tool for the forensic pathologist who approaches a corpse in an advanced state of decomposition. In fact, anomalies of decomposition, affecting the TBS, can easily occur because of ante mortem injuries or post mortem scavenging of human remains due to insects or larger animals such as wolfs, dogs, birds, etc. [64].

Regarding the entomological estimate, our study showed more than two-thirds of the estimations positively arranged the PMI derived from circumstantial information. In other cases, an underestimation of minPMI was found. This observation does not surprise—despite it is, very often, a matter of discussion in the court of law—due to the fact that with insects the minimum time since death is calculated. In fact, for estimating PMI from immature insects it has to be considered also the development interval and the pre-appearance interval (PAI), which is currently scientifically invaluable [65,66,67,68]. The “underestimation” is explained as the time of insects’ occurrence on a corpse. The casuistry considered for the study was composed of indoor cases, whereas the colonisation of bodies can record delays, depending on how easy it is for gravid females to access the body. Access of the body to insects is the second most important variable affecting the decomposition rate of the human body after temperature [9]. Since blowflies (Diptera: Calliphoridae) are usually the first necrophagous fauna to find a cadaver, the PMI estimation needs to take into account the factors that may delay the arrival of adult flies and subsequent oviposition by the females [68,69,70]. Further works, based on real cases have, however, to consider the time of colonization in indoor cases by other taxa (e.g.: Phoridae), and their informative potentiality, as already suggested by Bugelli and other authors [71,72]. The interclass correlation (0.71) calculated turned out to be “good” (very close to “excellent”). So, these results emphasise the applicability of forensic entomology in legal and medical cases, in accordance with the international guidelines and standards that help and strengthen forensic practitioners’ assessment. Nevertheless, these real forensic cases could be an important factor in improving data collection, and thus optimising scientific results and inferences.

Further retrospective studies and experimentation research should indeed be conducted using cases with a known and large PMI so that not only the precision of the estimation could be measured but also the accuracy. Moreover, other PMI estimation assessments could be taken into account.

## 5. Conclusions

In conclusion, forensic pathologists must consider every aspect before giving a statement about PMI. They always have to be aware of the possible source of errors and variability, particularly when dealing with cadavers in active-advanced decomposition. If the decomposition stage of the corpse is highly developed, additional pieces of information can be provided by ADD and entomological assessments of PMI.

In our study, the ADD method has shown some limitations, probably due to the experimental morphological approach still not being validated. The TBS and the derived ADD tend to overestimate the degree of the decomposition. The high variability can probably be due to the different stages of decomposition, especially for the head district, even if all the body came from the same geographical area and death occurred in indoor environment [29,73].

The results of our study, as observed by other authors in the literature [31,74], showed that scoring scales and regress equations derived for predicting ADD seem to be of little help in forensic practice, because of the so many factors affecting human decomposition and leading to irregular decomposition patterns. Furthermore, the TBS compared to ADD proved to be a good method for a PMI over 5 days. This emerging result represents an interesting new finding that still needs more research to be statistically validated and considered in real forensic practice.

On the other hand, forensic entomology has been confirmed as a well-validated approach in the field.

The study of these cases has therefore led to confirm that the PMI estimate of human remains requires multidisciplinary activity between different professional figures (anthropologists, entomologists, botanists, geologists and zoologists) aimed to integrate and interpret available data, all subjected to the medicolegal assessment of cadaveric decomposition.

## Figures and Tables

**Table 1 insects-12-00264-t001:** Summary of the circumstantial, environmental temperature and stage of decomposition for each case (TBS: Total Body Score).

Case	Month and Year of Discovery	Environmental Temperature (Average ± SD, °C)	Stage of Decomposition			
			Head/Neck Score	Trunk Score	Limbs Score	TBS
1	X.2016	16.5 ± 1.0	4	3	2	9
2	III.2017	8.0 ± 2.0	6	3	4	13
3	II.2017	7.5 ± 1.5	4	3	3	10
4	V.2017	20.5 ± 3.0	5	3	4	12
5	IV2017	14.0 ± 1.5	4	3	4	11
6	III.2017	8.5 ± 2.5	4	4	4	12
7	X.2017	14.5 ± 1.5	4	3	3	10
8	I.2018	6.0 ± 2.5	4	4	3	11
9	II.2018	4.0 ± 2.5	5	3	4	12
10	II.2018	5.5 ± 1.0	5	4	4	13
11	XII.2018	3.5 ± 2.5	5	4	4	13
12	VII.2016	28.5 ± 1.5	5	4	3	12
13	VIII.2016	28.5 ± 1.5	5	4	3	12
14	VIII.2017	28.5 ± 2.5	5	3	4	12
15	IV.2018	18.0 ± 2.5	5	4	3	12
16	VIII.2018	27.0 ± 2.0				
17	IV.2016	20.5 ± 1.0	4	3	2	9
18	X.2018	16.5 ± 2.0	8	3	3	14
19	VII2018	26.0 ± 2.5	6	4	3	13
20	X.2016	20.0 ± 4.0	5	3	3	11
21	VIII.2018	29.0 ± 1.0	6	4	3	13
22	VII.2018	29.0 ± 1.0	7	5	4	16
23	VII.2018	26.0 ± 2.0	4	3	3	10
24	VIII.2016	26.0 ± 20	5	4	4	13
25	VIII.2018	29.0± 0.5	5	5	3	13
26	VII.2018	28.0± 1.5	7	5	3	15
27	IV.2017	15.0 ± 3.0	2	3	2	7
28	VIII.2017	25.5 ± 2.0	6	5	4	15
29	VI.2017	28.5 ± 2.5	5	3	4	12
30	VII.2018	26.5 ± 1.5	6	4	3	13

**Table 2 insects-12-00264-t002:** Comparison between the circumstantial data estimation and technical assessment using accumulated degree-days (ADDs) converted in a 24 h range.

Taxon	Case Number	PMI from Circumstantial Data		TBS PMI Estimation		Entomological min PMI Estimation	
*C. vicina*	1	120	144	120	144	120	270
	2	168	192	168	192	120	200
	3	120	144	120	144	96	170
	4	120	144	120	144	96	180
	5	144	168	144	168	120	180
	6	144	168	144	168	96	144
	7	96	120	144	168	96	170
	8	96	120	144	168	96	170
	9	144	168	144	168	120	200
	10	96	120	168	192	72	110
	11	168	192	168	192	110	180
*L. sericata*	12	72	96	96	120	48	72
	13	72	96	96	120	48	72
	14	72	96	96	120	48	72
	15	168	192	168	192	84	180
	16	72	96	-	-	48	84
	17	96	120	96	120	78	108
	18	192	216	240	264	132	204
	19	96	120	168	192	54	90
	20	168	192	168	192	96	216
	21	48	72	72	96	48	54
	22	48	72	72	96	48	54
	23	216	240	96	120	48	84
	24	216	240	96	120	54	84
	25	48	72	144	168	48	60
	26	48	72	144	168	48	60
	27	144	168	120	144	168	204
*C. albiceps*	28	144	168	144	168	84	144
	29	72	96	96	120	60	108
	30	96	120	120	144	60	102

(green = positive concordance; yellow = concordance at ranges’ limits; orange = no concordance) (PMI: post mortem interval; TBS: TotalBody Score).

## Data Availability

Data are available upon request from the authors.

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
