# Peer review of "Comparison of Accumulated Degree-Days and Entomological Approaches in Post Mortem Interval Estimation"

_insects, 2021, doi:10.3390/insects12030264_

Round 1

Reviewer 1 Report

An excellent paper, unique in content and topically relevant - deserving immediate publication, clearly written and following sound logic. I have but two minor comments:

Line 69: change ‘methodologies’ to ‘methods’. Methodology is the study of methods, which is not the correct context in this sentence.

Line 262: insert ‘being’ - thus: morphological approach still not being validated.

Well done!

Andrew Whittington

Author Response

An excellent paper, unique in content and topically relevant - deserving immediate publication, clearly written and following sound logic.

Reply: Dear Andrew thank you indeed for your comments and appreciation

I have but two minor comments:

Line 69: change ‘methodologies’ to ‘methods’. Methodology is the study of methods, which is not the correct context in this sentence.

Reply: Done

Line 262: insert ‘being’ - thus: morphological approach still not being validated.

Reply: Done

Reviewer 2 Report

Line 21, add a comma after "Among them"

Line 84.  reword to "A few authors have tested Megyesi's research in their countries and have found  positive correlation".

This is a well written paper, which provides insight to a question that now frequently asked:  is entomology or TBS best, and how to both compare with corporal evidence.  However, the authors should re-write the section materials and methods that addressed entomology to improve the reader comprehension of their research intent.  The authors discuss the collection method, and the identification method.   They even discuss the ADD data obtained from average temperatures.  However, the do not discuss that most of the development data in the literature for insects is based on ADD accumulation.  I am familiar with their reference [58].  The methods of the paper would be more clear if the authors discussed the issues presented with ADD calculations as the foundation of most larval development, and the ADD used in the TBS system.  Discussion of how they accounted for this issue in existing literature would benefit the article.

Additionally, Table 2 is titled with "ADD PMI Estimation" and "Entomological PMI estimation".  Perhaps the the "ADD PMI estimation" title be changed to "TBS PMI Estimation" to reduce confusion on the use of ADD as a foundation for much of the PMI estimation work in forensic entomology. 

Author Response

Line 21, add a comma after "Among them"

Reply: Done

Line 84.  reword to "A few authors have tested Megyesi's research in their countries and have found  positive correlation".

Reply: Done

This is a well written paper, which provides insight to a question that now frequently asked:  is entomology or TBS best, and how to both compare with corporal evidence.  However, the authors should re-write the section materials and methods that addressed entomology to improve the reader comprehension of their research intent.  The authors discuss the collection method, and the identification method.   They even discuss the ADD data obtained from average temperatures.  However, the do not discuss that most of the development data in the literature for insects is based on ADD accumulation.  I am familiar with their reference [58].  The methods of the paper would be more clear if the authors discussed the issues presented with ADD calculations as the foundation of most larval development, and the ADD used in the TBS system.  Discussion of how they accounted for this issue in existing literature would benefit the article.

Reply:  Paragraph 2.3 has been amended, literature example are reported and the references updated as in the following lines:

2.3. PMI estimation and statistical analysis

Insect development is temperature dependent and species/population specific in the range between the minimal and the maximal developmental thresholds. Depending on the nature of the sample (already fixed or living specimens, larvae, close or empty puparia) two different methods can be applied to estimate the age of an insect. Currently ADD is the most common method used for this estimation when living specimens can be bread till the adult stage or when empty puparia are collected from the crime scene. Because of the minimal developmental threshold, the ADDs used in entomology differ from the ADDs applied in the anthropological method. For this reason, ADDs were derived from the average temperature as absolute value for the morphological evaluation, whereas if needed for the entomological evaluation the minimal developmental threshold was subtracted from the absolute value.

minPMI was evaluated also using the larvae measurements applying the data available in literature [58] when the samples were already fixed. In this case the size of the larvae is compared with diagrams reporting the relations between size, temperature and time. Whereas some authors report only the size as function of the temperature and time [58-60], others provide also the formulas and the interval of confidence of the larval length based on the ADDs [61]. Larvae were counted and then measured using Leica M60 stereomicroscope equipped with CCD camera (Wetzlar, Germany) and the automatic measurement tool. When numbers of larvae were under fifty, measurements were performed on all the specimens. When the count exceeded fifty, only fifty larvae were considered. Measurements were expressed in mm as average and standard deviation (SD).

The estimations derived from the two methods were compared using an interclass correlation according to Cicchetti (1994) [62] and Koo and Li (2016) [63]. Statistical analyses were performed using the SPSS software.

Additionally, Table 2 is titled with "ADD PMI Estimation" and "Entomological PMI estimation".  Perhaps the the "ADD PMI estimation" title be changed to "TBS PMI Estimation" to reduce confusion on the use of ADD as a foundation for much of the PMI estimation work in forensic entomology. 

Reply: Thank you for the suggestion, done